# Updates on the Management of Acute Myeloid Leukemia

**DOI:** 10.3390/cancers14194756

**Published:** 2022-09-29

**Authors:** Sofía Huerga-Domínguez, Sara Villar, Felipe Prósper, Ana Alfonso-Piérola

**Affiliations:** Hematology and Hemotherapy Department, Clínica Universidad de Navarra, 31008 Pamplona, Spain

**Keywords:** acute myeloblastic leukemia, target-therapy, gemtuzumab ozogamicin, midostaurin, vyxeos, venetoclax, ivosidenib, CC-486, magrolimab, quizartinib

## Abstract

**Simple Summary:**

Acute myeloid leukemia is the most common type of acute leukemia in adults. It is associated with poor outcomes, especially in older patients. Treatments based exclusively on chemotherapy do not achieve high overall survival rates, or in any case, only in a small group of patients. The objective of this review is to discuss the treatment of acute myeloid leukemia from newly approved therapeutic drugs to newer strategies. In first line, the combination of new targeted therapies with standard chemotherapy achieves better outcomes in “fit” patients. For “unfit” patients the combination of different targeted therapies provides them with better overall survival rates, with limited toxicity. As for refractory and relapsed acute myeloid leukemia, development of immunotherapy or new targeted therapies brings new hope.

**Abstract:**

Acute myeloid leukemia is a heterogeneous disease defined by a large spectrum of genetic aberrations that are potential therapeutic targets. New targeted therapies have changed the landscape for a disease with poor outcomes. They are more effective than standard chemotherapy with a good safety profile. For “fit patients” in first-line, the combination of gemtuzumab ozogamicin or midostaurin with intensive chemotherapy or Vyxeos is now considered the “standard of care” for selected patients. On the other hand, for “unfit patients”, azacitidine-venetoclax has been consolidated as a frontline treatment, while other combinations with magrolimab or ivosidenib are in development. Nevertheless, global survival results, especially in relapsed or refractory patients, remain unfavorable. New immunotherapies or targeted therapies, such as Menin inhibitors or sabatolimab, represent an opportunity in this situation. Future directions will probably come from combinations of different targeted therapies (“triplets”) and maintenance strategies guided by measurable residual disease.

## 1. Introduction

Acute myeloid leukemia (AML) is the most common type of acute leukemia in adults, and it is associated with poor outcomes, especially in older patients. It is a heterogeneous disease, defined by a broad spectrum of cytogenetic and molecular aberrations. These aberrations are used to divide patients into prognostic subgroups and are conveniently potential therapeutic targets as well [1,2].

The advances in basic and translational investigation and the development of new target therapies have changed the treatment of AML, with US Food and Drug Administration (FDA) and European Medicines Agency (EMA) approvals for several new treatments in recent years. Nevertheless, primary refractory, relapsed, high-risk patients who are unfit for allogeneic stem cell transplantation (alloSCT) still do not have effective therapeutic alternatives. Hence many investigational drugs are currently being explored [3].

One of the most important challenges in the treatment of AML lies in assessing patient fitness to receive intensive treatments with curative intent. This evaluation should include an integral evaluation of the patient, with clinical characteristics (age, performance status, comorbidities, quality of life, patient’s preferences and family support) evaluable through different geriatric scores and biological factors of the disease. Intensive chemotherapy with attenuated doses can be considered in frail patients with a high probability of long-term remission, high chemosensitivity and a favorable cytogenetic or molecular risk [2,4].

The objective of this review is to discuss AML treatment in first-line and refractory or relapsed patients from newly approved therapeutic drugs to newer strategies, such as mutation-driven targeted treatment combinations or novel immunotherapies.

## 2. First-Line Treatments

### 2.1. First-Line Treatments for “Fit” AML Patients

For many years, the only therapeutic alternative for newly diagnosed AML “fit” patients was intensive chemotherapy based on cytarabine + anthracycline (“7 + 3”). In addition, allogeneic stem cell transplantation played an important role in improving the survival of eligible intermediate- or high-risk AML patients. Targeted therapies combined with standard chemotherapy or new formulations of older drugs have expanded the treatment options and improved survival [5].

#### 2.1.1. Gemtuzumab Ozogamicin (GO)

Gemtuzumab ozogamicin (GO) is a combination of the chemotherapy agent calicheamicin, and a recombinant humanized IgG4 antibody directed against CD33 [1]. CD33 is a suitable target in AML because it is highly and selectively expressed in most AML blasts, with some expression in hepatocytes [6].

The history of GO dates to 2000, when it was approved as a treatment for older refractory or relapsed CD33+ AML patients (doses 9 mg/m^2^/day). Nevertheless, further contradictory data in terms of efficacy and mostly toxicity meant it was retired in 2010 [2].

However, in 2012, the ALFA-0701 trial, a phase III randomized trial, showed that the use of fractionated lower doses of GO (3 mg/m^2^ on days 1, 4, 7) allowed a safe delivery of higher cumulative doses if the single dosage did not exceed 3 mg/m^2^. The addition of GO to induction and consolidation achieved a significant improvement in event-free survival (EFS) and overall survival (OS) (2-year OS 53.2% vs. 41.9% *p* = 0.037) with no increase in fatal toxicities. Nevertheless, the benefit associated with GO was not apparent in patients with unfavorable cytogenetics (HR 0.59 (0.32–1.09), *p* = 0.31). No correlation between CD33 expression and survival endpoint was observed [7]. In the phase III AML15 study by the UK`s Medical Research Council, a significant survival benefit was associated with a single dose of GO (3 mg/m^2^) during the first and third courses of chemotherapy in patients with favorable cytogenetics (OS 79% vs. 51%; *p* = 0.001) [8]. Further studies did not show an improvement in any survival endpoint of GO as maintenance therapy [9,10].

Since the association between high CD33 expression and the NPM1 mutation is well known, the AMLG 09-09 Phase III Study explored the effect of GO added to induction and first consolidation therapy in adult patients with NPM1-mutated AML. EFS was not significantly different between the two arms due to the higher early mortality rate in the GO arm in older patients and the better outcome overall in both arms [11]. However, GO led to a better reduction in NPM1 mutant transcript levels and significantly reduced the cumulative incidence of relapse (CIR) (4-year CIR rate 31.6% vs. 43.9%, *p* = 0.015) [12].

Hills et al. confirmed in a later meta-analysis the results of the ALFA-0701 trial and AML15 study. The benefit of GO depended predominantly on those patients with favorable (absolute survival benefit at 6 years 20.7% OR 0.47 (0.31–0.73) *p* = 0.0006) and intermediate cytogenetics (absolute survival benefit at 6 years 5.7%, OR 0.84 (0.75–0.95) *p* = 0.005) while patients with adverse cytogenetic characteristics did not benefit (2.2%, OR 0.99 (0.83–1.18) *p* = 0.9) [13].

The main adverse events are veno-occlusive disease (VOD) and thrombocytopenia. However, the most recent trials did not suggest that the VOD risk is higher than the benefit of GO if patients are selected, and the GO unique dose is lower than 6 mg/m^2^ [6]. Prophylaxis with defibrotide in alloSCT after GO could be considered.

To sum up, the combination of GO with induction and consolidation chemotherapy is safe and improves outcomes in patients with de novo CD33+ AML when administered in fractioned doses of 3 mg/m^2^ (days 1, 4, and 7) or alternatively as a single dose (3 mg/m^2^) on day 1. Taking the previous data into account GO should be reserved for patients with favorable or intermediate cytogenetics and no high risk of VOD.

#### 2.1.2. Vyxeos (CPX-351)

Vyxeos (CPX-351) is a liposomal formulation of cytarabine and daunorubicin in a fixed 5:1 molar ratio, designed to deliver synergistic drug ratios to leukemia cells and reduce toxicity. It is currently the first-choice treatment for “fit”, newly diagnosed, secondary AML (s-AML), therapy-related AML (tAML) or AML with myelodysplasia-related changes (AML-MRC) [14].

Vyxeos was compared to 7 + 3 induction in a phase III trial for 60- to 75-year-old patients with newly diagnosed secondary AML. The rates of OS (9.5 vs. 5.9 months), EFS (2.53 vs. 1.31 months) and complete response (CR) (47.7% vs. 33.3%) were higher in the CPX-351 group as compared to standard induction. Of note, 34% of patients in the CPX-351 arm underwent allogeneic alloSCT, compared to 25% in the 7 + 3 arm. An exploratory landmark survival analysis from the time of alloSCT favored CPX-351 (*p* = 0.009) [15].

These results have been extended to a real-life setting in a French study in 2021, which included 103 patients treated with Vyxeos. The overall response rate (ORR) after induction was 59%, and minimal residual disease (MRD) <10^−3^ was achieved in 57% of CR patients, with an OS of 16.1 months at a median follow-up time of 8.6 months (MRD was assessed by flow cytometry, NGS (next-generation sequencing) or real-time quantitative polymerase chain reaction according to the standard MRD methodology used at the site. All patients with MRD <10^−3^ were considered negative). In addition, CPX-351 improves the poor prognosis associated with *ASXL1* and *RUNX1* mutations but not in *TP53* mutated patients [16].

The safety profile of CPX-351 was similar to that of conventional 7 + 3 therapy, despite a prolonged time to neutrophil and platelet count recovery with CPX-351 (36.5 vs. 29 days) [15].

It remains to be evaluated if younger patients with secondary AML would also benefit from CPX-351. In the EHA 2022 meeting, the latest data for the AML19 trial were presented. It randomized CPX-351 vs. FLAG-Ida in young (<60 years) patients with high-risk AML or myelodysplastic syndromes (MDS) (>10% blasts). CPX-351 did not improve OS compared to FLAG-Ida but was associated with better relapse-free survival (RFS) (mRFS 22.1 vs. 14 months, *p* = 0.03) and duration of remission (319.5 vs. 167 days, *p* = 0.046) [17].

#### 2.1.3. FLT3 Inhibitors: Midostaurin and Others

Midostaurin is an oral first-generation multitargeted kinase inhibitor. Combined with standard chemotherapy, it is currently the first choice treatment for “fit”, newly diagnosed, FLT3-mutated AML patients [1].

Its approval was based on the RATIFY trial. In this study, patients were randomly assigned to receive induction and consolidation standard therapy plus either midostaurin or a placebo. Subsequently, those who were in remission after consolidation therapy entered a maintenance phase with midostaurin or the placebo. The addition of midostaurin to standard chemotherapy significantly improved OS and EFS when compared with placebo (4-year OS 51.4% vs. 44.3%) [18].

Maintenance treatment with midostaurin was well tolerated, but analyses did not prove better outcomes in terms of OS or EFS; nevertheless, the trial was not designed to determine the independent effect of a maintenance phase [19]. However, the RADIUS trial demonstrated that midostaurin maintenance therapy post-alloSCT might be a viable option to reduce the risk of relapse in some patients. The estimated 18-month RFS was 89% (69–96%) in the midostaurin arm and 76% (54–88%) in the standard-of-care arm (HR 0.46 (95% CI (0.12–1.86) *p* = 0.27). Despite an improvement in OS with the addition of midostaurin (estimated 24-month OS (95% CI) 85% (65–94%) versus 76% (54–89%) (HR, 0.58 [95% CI, 0.19–1.79]; *p* = 0.34), the results were not statistically significant). [20]. Therefore, the benefit of post-remission maintenance strategies using midostaurin or any other targeted agents is still under investigation.

The RADIUS-X expanded access program confirmed the manageable safety profile of midostaurin. In addition to adverse events that are typically associated with intensive chemotherapy, no unexpected adverse events were observed [21].

On the other hand, there are other FLT3 inhibitors that are being studied in the first-line. Data from the phase III QuANTUM-First trial were presented at the EHA 2022 meeting. It enrolled newly diagnosed FLT3-ITD mutated AML patients (FLT3-TKD mutations were excluded) who were randomly assigned to receive treatment with Quizartinib, a more specific and potent FLT3 s-generation inhibitor, or placebo combined with standard induction chemotherapy. Patients who achieved CR/CRi (complete remission with incomplete blood count recovery) could receive up to four cycles of consolidation chemotherapy, plus Quizartinib and Quizartinib, which were continued for up to 3 years following consolidation therapy and/or alloSCT. The median OS (31.9 vs. 15.1 months) and RFS (HR 0.733; 95% CI, 0.55–0.97) were significantly better in the Quizartinib arm vs. standard of care. It had a manageable safety profile, with no addition of toxicity [22].

#### 2.1.4. Other Drugs

There are new strategies based on combinations of other targeted therapies with standard chemotherapy under study:
Venetoclax (VEN) (bcl-2 inhibitor) has an emerging role in “unfit” AML patients. The CAVEAT study, a phase Ib dose-escalation study, showed that VEN combined with 5 + 2 induction chemotherapy (2 days idarubicin per 5 days cytarabine) was safe in fit, older (>65 years) patients. The high remission rate in de novo AML (ORR 97%; CR 68%, CRi 19%) warrants additional investigations [23] (NCT04070768 Venetoclax + GO; NCT04038437 Venetoclax + Vyxeos)Ivosidenib and enasidenib (IDH1/2 inhibitors), in combination with intensive induction and consolidation therapy, were well tolerated in patients with newly diagnosed IDH1 or IDH2 AML. The initial clinical activity was encouraging (ORR 77% in the ivosidenib-treated cohort and 63% in the enasidenib-treated cohort), and the benefit of adding these drugs to standard chemotherapy is being further evaluated in a phase 3 trial (HOVON150AML NCT03839771) [24].

### 2.2. First-Line Treatments for “Unfit” AML Patients

Older patients with AML are unfit for intensive therapies and have a dismal prognosis. The therapeutic challenges are, on the one hand, the more aggressive biology of the disease and the poor performance status and comorbidities [25]. However, the hypomethylating agents (HMA), azacitidine and decitabine in monotherapy have been the “de facto” standard of care for them. Nevertheless, the median OS for patients treated with azacitidine (AZA) was 10.4 months in the pivotal international phase III trial [26] and 7.1 months in a large population-based study in the United States [27]. Considering these poor outcomes, combinations of HMA with other drugs have been developed.

#### 2.2.1. Venetoclax

VEN is an oral, highly selective B-cell lymphoma 2 (BCL2) inhibitor. It induces apoptosis in AML blasts that are dependent on BCL2 for survival. High expression of BCL2 is associated with an inferior response to intensive chemotherapy and significantly shorter survival (*p* < 0.005) [28]. The FDA and EMA have approved VEN for newly diagnosed AML in patients ≥75 years or ineligible for intensive chemotherapy in combination with HMA or low-dose cytarabine (LDAC) [25].

In the pivotal phase III randomized trial (VIALE-A), untreated patients with AML ≥75 years or who were ineligible for standard induction were randomly assigned to AZA (75 mg/m^2^ on days 1–7) plus either VEN (target dose 400 mg once daily) or placebo. At a median follow-up of 20.5 months, the median OS was 14.7 in the AZA-VEN group vs. 9.6 months in the control group (HR 0.66, *p* < 0.001). Moreover, CR rate was also higher in the VEN arm (36.7% vs. 17,9%, *p* < 0.001) [29].

More recent data suggests that IDH1/2 mutated AML benefitted better from the combination (ORR 72 vs. 60%, mOS 24.5 vs. 12.3) [30], while monocytic AML is more resistant because it loses expression of BCL2 [31].

The main adverse events included nausea and grade 3 or higher neutropenia and thrombocytopenia. The combination of AZA and VEN was also associated with a higher incidence of febrile neutropenia (42% vs. 19%). There are recommendations in terms of dose adjustments in case of hematologic toxicity [29].

The VIALE-C study was a phase 3 randomized trial that compared VEN versus placebo in combination with LDAC in patients unfit for intensive therapy. The study failed to achieve its primary endpoint (improved OS), although an unplanned analysis with an additional six months of follow-up showed a significantly longer median OS (8.4 vs. 4.1 months), CR/CRi (56% and 28% vs. 16% and 10%) and EFS rates in the VEN-LDAC arm [32].

Unfortunately, patients who failed VEN + HMA display high-risk disease biology and particularly poor survival, with a median OS of only 2.4 months [33]. Resistance to VEN-based combinations is related to the acquisition of clones activating signaling pathways such as *FLT3*, *RAS*, or biallelically *TP53* loss [34].

#### 2.2.2. Glasdegib

Glasdegib is a selective oral inhibitor of the Hedgehog pathway. Combined with LDAC, it is indicated for treating newly-diagnosed AML patients unfit for standard chemotherapy.

Its approval was based on a phase II randomized, open-label, multicenter study that evaluated the efficacy of glasdegib plus LDAC in patients with AML or high-risk myelodysplastic syndrome unsuitable for intensive chemotherapy. The median OS was 8.8 versus 4.9 months. The main nonhematologic adverse events were pneumonia (16.7%) and fatigue [35]. There are no studies that compare LDAC + Venetoclax vs. LDAC + Glasdegib.

#### 2.2.3. Ivosidenib

Ivosidenib (IVO) is an oral, targeted small-molecular inhibitor of mutant *IDH-1*. Mutant *IDH-1* catalyzes the production of D-2-hydroxyglutarate, favoring epigenetic dysregulation and oncogenesis. *IDH-1* mutated AML (6–10%) is associated with older age and a poor prognosis, especially with a normal karyotype. IVO combined with AZA showed clinical activity with a safety profile in a phase 1b trial in 2019 [36]. The AGILE trial, a global, double-blind, phase 3 trial, randomly assigned patients with newly diagnosed *IDH-1* mutated AML who were ineligible for intensive induction to receive AZA plus either IVO (500 mg once daily) or placebo. At a median follow-up of 12.4 months, EFS (HR treatment failure, relapse from remission, or death, 0.33; 95% CI 0.16–0.69; *p* = 0.002) and OS (24 vs. 7.9 months) were significantly longer in the IVO-AZA arm (*p* = 0.001). The safety profile was favorable, with fewer infections reported in the IVO group (28% vs. 49%) and an incidence of differentiation syndrome of only 14% [37]. Despite mOS being seemingly superior in the AGILE trial (mOS 24 months) than in studies that evaluate AZA-VEN in unfit *IDH-1* mutated AML (mOS 21.9 months) [30], there are no studies comparing, front to front, these two regimens.

Recently, this combination was approved in the United States for newly diagnosed *IDH1*-mutated AML patients who are ≥75 years old or who have comorbidities that preclude the use of standard chemotherapy.

#### 2.2.4. Other Drugs

Furthermore, there are various combinations of hypomethylating agents with other drugs currently undergoing evaluation in early clinical trials.
Magrolimab: magrolimab is a monoclonal anti-CD47 antibody. CD47 is a “do not eat me” signal, overexpressed in myeloid malignancies, that avoids the tumor cells’ phagocytosis. Blockade of CD47 causes inhibition of the negative phagocytic signal and induces the elimination of leukemic stem cells. Magrolimab has been granted Orphan Drug Designation by the FDA for MDS [38].The combination of magrolimab and AZA showed robust activity in a phase 1b clinical trial in 2019. It demonstrated exciting results in MDS and in untreated AML patients (CR/CRi 56%), being particularly effective in *TP53*-mutant AML, a treatment-refractory subgroup [39].Thus, in the EHA 2022 meeting, results for the phase 1b trial that evaluated the tolerability and efficacy of Magrolimab combined with AZA in high-risk, newly diagnosed, *TP53*-mutated AML unfit for intensive chemotherapy were presented. It showed durable responses (41.6% CR/CRi with a median duration of 7.7 months) and a median OS of 10.8 months with a manageable safety profile. The *TP53* VAF (variant allele frequency) was reduced to <0.07 by cycle 5 (day 1) in 7/9 longitudinally evaluable responders. Magrolimab + AZA is currently being studied in patients with frontline *TP53*-mutated AML in a phase 3 trial (ENHANCE-2; NCT04778397) [40].APR-246 (Eprenetapopt): APR-246 is a novel, small molecule that induces apoptosis in *TP53* mutated cancer cells. It stabilizes the TP53 protein and restores its function as a tumor suppressor gene [41]. *TP53* mutation, especially in a multi-hit state, has been associated with a complex karyotype, secondary AML, high-risk presentation, and poor outcomes and used to be considered “undruggable” in the past [42]. APR-246 has been granted Orphan Drug Designation by the FDA for MDS.A phase Ib/II study has combined APR-246 with AZA in *TP53*-mutated MDS or AML with encouraging results (ORR 64% with a CR rate of 36% and a median OS of 10.8 months) [43].Finally, the preliminary results of the VEN-A-QUI trial were also shown at the EHA 2022 meeting. It was a phase I/II trial that assessed the safety and efficacy of the combination of AZA or LDAC with VEN and Quizartinib in newly-diagnosed AML patients aged ≥60 years old unfit for intensive treatment (59% secondary AML, 48% exposed to AZA before). It achieved an ORR of 54%; however, substantial toxicity was observed [44]. Deeper studies are necessary to determine the efficacy and security of these triplets.

## 3. Maintenance

Once the induction and consolidation treatment has been completed, disease relapse remains the most common cause of death, even after alloSCT. As we just mentioned, a maintenance treatment that would prolong the remission duration without excessive toxicity is an area without any strong evidence-based option.

The first therapy approved in the maintenance setting is CC-486. CC-486 is an oral formulation of AZA, but its pharmacokinetic and pharmacodynamic profiles are distinct from those of injectable AZA and cannot be used interchangeably. In fact, it showed efficacy in patients who previously developed resistance to injectable HMA [45,46]. It is approved as maintenance therapy for AML patients in CR/CRi after intensive induction with/without consolidation therapy who are not candidates for alloSCT, based on the results of phase III, double-blind QUAZAR AML-001 trial. This trial included AML patients (55 to 86 years) in first CR after intensive chemotherapy who were not candidates for alloSCT that were randomly assigned to receive either CC-486 (once daily for 14 days per 28-day cycle) or placebo. Median OS and EFS were significantly longer with CC-486 as compared to the placebo (OS 24.7 vs. 14.8 months, *p* < 0.001) [47]. Multivariate analyses showed that CC-486 significantly improved OS and RFS independent of baseline MRD status and resulted in a higher rate of conversion from MRD+ to MRD− (37% vs. 19%) [48]. The most common adverse events are gastrointestinal ones and neutropenia [47].

In addition, a phase II study suggested that oral azacitidine may provide effective maintenance therapy after alloSCT, with low rates of relapse, disease progression (1-year rate of relapse was 21%), and GVHD (graft-versus-host disease; 30% chronic GVHD); but more studies are necessary [49,50].

Furthermore, in the context of post-alloSCT treatment, the maintenance with Sorafenib, a multitargeted tyrosine kinase inhibitor, has proved to reduce the risk of relapse and death in FLT3-ITD-positive AML patients in CR (HR relapse or death 0.39, log-rank *p* = 0.013) [51].

## 4. Refractory and Relapsed AML

The prognosis of relapsed and refractory AML (R/R AML) remains poor (median OS around 7 months), with no satisfactory or standard salvage treatment [52]. The knowledge of the molecular pathophysiology of AML has allowed us to identify novel driver mutations that have become objectives for new promising therapeutic alternatives [53]. Due to clonal evolution, it is advisable to repeat mutational profiling at the time of relapse in order to select the best-targeted therapy [1].

### 4.1. FLT3 Inhibitors: Gilteritinib

Gilteritinib is an oral, second-generation, and highly selective *FLT3* inhibitor [54]. It is currently approved by FDA and EMA for the treatment of relapsed or refractory *FLT3-*mutated AML.

Its approval was based on the ADMIRAL trial, a randomized phase III trial where patients with relapsed or refractory *FLT3*-mutated AML were randomly assigned to receive either gilteritinib (120 mg per day) or salvage chemotherapy according to local investigators’ choice. The median OS and EFS were significantly longer in the gilteritinib group (mOS 9.3 vs. 5.6 months; *p* < 0.001). It was associated with higher CR/CRi rates, too (34% vs. 15.4%). Noticeably, the adverse events of grade 3 or higher were less frequent in the gilteritinib group [55].

Another study analyzed the potential impact of prior *FLT3*-TKI exposition (midostaurin or Sorafenib) in clinical outcomes of relapsed/refractory patients treated with gilteritinib. The CR rates were 58% with an OS of 7.8 months, suggesting that gilteritinib is not less effective after previous tyrosine kinase inhibitors (TKI) treatment [56].

### 4.2. IDH 1/2 Inhibitors

#### 4.2.1. Enasidenib

Enasidenib is an oral, selective inhibitor of mutant *IDH2*. It is approved by the FDA (not EMA) for relapsed/refractory *IDH2*-mutated AML. The approval was based on a phase I/II trial, in which enasidenib as a single agent (100 mg daily) achieved an ORR of 40.3% with a median duration of response of 5.8 months and a median OS of 9.3 months in R/R *IDH2*-mutated AML patients. It was well tolerated with an incidence of differentiation syndrome of only 7% [57].

#### 4.2.2. Ivosidenib

In addition to its ever-growing role in first-line treatment, IVO was approved in 2019 for R/R AML in patients unfit for intensive chemotherapy. The results of the phase I dose-escalation study were comparable to Enasidenib. IVO as a single agent (500 mg daily) showed an ORR of 41.6%, with a median duration of response of 8.2 months in R/R IDH1-mutated AML patients [58].

Nevertheless, the phase III IDHENTIFY study evaluating enasidenib versus conventional care regimens failed to meet the primary endpoint of OS. *IDH* inhibitors alone are unlikely to provide durable remissions, but they could be an alternative to control the disease with low toxicity rates or as a bridge to alloSCT [3].

### 4.3. TP53 Inhibitors: Idasanutlin (MDM2 Inhibitor)

Idasanutlin is a selective, small-molecule MDM2 antagonist. MDM2 is a primary negative regulator of *TP53* and can be overexpressed in human tumors, including AML. An MDM2-targeted therapy could stabilize *TP53*, activate its tumor-suppressor function, and promote apoptosis [59]. Idasanutlin, alone or in combination with cytarabine, demonstrated tolerable safety and clinical activity in a phase I/Ib trial for R/R AML [60].

These results have encouraged the recently published MIRROS trial, a multicenter, phase 3 trial evaluating cytarabine 1 g/m^2^ in combination with idasanutlin (300 mg twice daily) vs. placebo in patients with R/R AML. Despite improved ORR (38.8% vs. 22%; OR 2.25 95% CI 1.36–3.72), idasanutlin plus cytarabine did not meet the primary endpoint, OS [61].

### 4.4. Monoclonal Antibodies: Gemtuzumab Ozogamicin (GO)

The FDA, but not the EMA, also approved GO as a single agent at a dosage of 3 mg/m^2^ for the treatment of patients with relapsed or refractory CD33^+^ AML [5].

### 4.5. Emavusertib (CS-4948)

Emavusertib (CA-4948) is an oral inhibitor of interleukin-1 receptor-associated kinase 4 (IRAK4) and *FLT3.* IRAK4 participates in inflammation and oncogenesis. Genetic mutations in the splicing factors *SF3B1* and *U2AF1* drive overexpression of a highly active isoform of IRAK4 and have been associated with poor prognosis in AML. Data from the phase I dose escalation trial were presented at the EHA 2022 meeting, suggesting Emasuvertib in monotherapy is well tolerated and effective in heavily pretreated AML, especially in those with spliceasome mutations (ORR 40%). Phases Ib and IIa are ongoing (NCT04278768) [62].

### 4.6. Uproleselan

Uproleselan is a novel E-selectin antagonist that enhances chemotherapy response and decreases chemotherapy toxicity in vivo. A phase I/II trial evaluated the safety and antileukemic activity of uproleselan (5–20 mg/kg) with MEC (mitoxantrone, etoposide, and cytarabine) among patients with R/R AML. The remission rate was high (41%), and the median OS was 8.8 months. In terms of toxicity, uproleselan was associated with low rates of oral mucositis [63].

### 4.7. MENIN Inhibitors

The interaction between menin and MLL-1 protein (mixed lineage leukemia 1) plays an important role in AML with KMT2A rearranged and mutated *NPM1*. The inhibition of the menin-MLL complex can inhibit proliferation and induce differentiation in these subtypes [64]. In addition, MLL1 translocations and *NPM1* mutations are frequently associated with other mutations, such as *FLT3*. Co-inhibition of menin and *FLT3* demonstrated enhanced antileukemia activity in *MLL-R*/*FLT3-*mutated and *NPM1*-mutated/*FLT3*-mutated AML [65]. Menin inhibitors and their combinations are currently being studied in clinical trials (NCT04067336, NCT04811560).

## 5. Other New Immunotherapies

Immunotherapy in AML treatment is not new; the graft versus leukemia effect and donor lymphocyte infusion in alloSCT is well-documented. Nevertheless, in the last few years, new immunotherapies, such as CAR-T cells or checkpoint inhibitors, have grown at a slower pace compared to other pathologies like solid tumors, ALL (lymphoblastic leukemia) or lymphomas [66].

### 5.1. TIM-3 Inhibitor: Sabatolimab

Sabatolimab (MBG453) is a humanized anti-TIM3 IgG4 antibody. T-cell immunoglobulin and mucin domain-containing 3 (TIM-3) is an immune checkpoint and a negative regulator of T cells. It is aberrantly overexpressed in AML stem cells and leukemic blasts [67].

Sabatolimab, in combination with HMA, in patients with both high-risk MDS and newly-diagnosed unfit AML was assessed in a phase I trial. In AML patients, the ORR rate was 40% (30% CR/CRi), with a median duration of response of 12.6 months. Later trials are ongoing to evaluate the combination of sabatolimab with AZA and VEN (STIMULUS-AML1 NCT04150029) [68].

### 5.2. Other Monoclonal Antibodies

Flotetuzumab is a dual-affinity re-targeting inhibitor (CD123 × CD3). In a phase I/II study for R/R AML patients, it showed an ORR rate of 30% and a median OS of 10.2 months. The most frequent adverse event was cytokine release syndrome, largely grades 1–2 [69]. It represents an innovative approach for R/R AML.

On the other hand, Cusatuzumab is a human antiCD-70 monoclonal antibody. In combination with AZA, it has demonstrated activity in previously untreated older patients (CR/CRi rate 83%) [70].

### 5.3. CAR-T Cells

CAR-T cells treatment for myeloid malignancies had been challenging due to the lack of AML-specific surface antigens with low “off-tumor” toxicity [71]. Despite strategies, like suicidal control of CAR-T cells, or temporary expression of the CAR, very limited clinical data have been presented, mainly CD33 and CD123 CAR-T cells [72]. The major limitation of these targets is hematologic toxicity, which requires consolidation with alloSCT in most cases. Nevertheless, new targets like CLL-1, CD70, NKG2D, more specific to LSC than HSC are being explored.

## 6. Conclusions

The AML treatment has changed in recent years. In first-line, for “fit” patients, the combinations of new drugs (Midostaurin, Quizartinib, Gemtuzumab) with chemotherapy are the best example of “target-therapy”. In the same token, for “unfit” patients, in addition to the consolidation of azacitidine-venetoclax as frontline treatment, we consider the association with other new therapeutic strategies like Magrolimab, Ivosidenib or APR246 (Table 1).

Nevertheless, despite all these recent therapeutic advances, global survival results, especially in R/R patients remain poor. Future directions will probably come from combinations of different targeted therapies (triplets), development of immunotherapy, maintenance strategies and establishing measurable residual disease as a key factor in therapeutic decision-making [73]. The projected roadmap for AML treatment in the future is detailed in Figure 1.

## Figures and Tables

**Figure 1 cancers-14-04756-f001:**
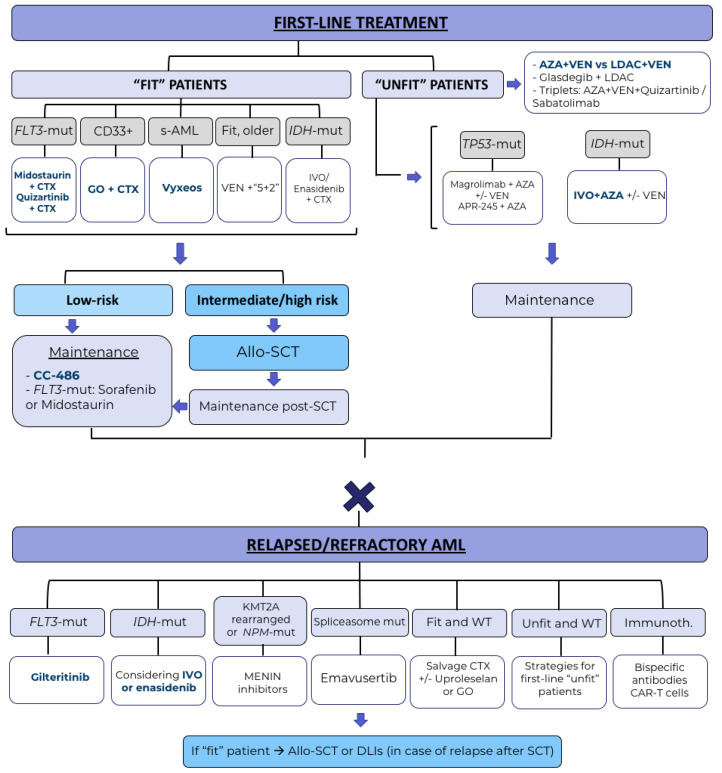
Projected roadmap for AML treatment in, the future. Treatment-combinations with phase III data are represented in blue. AZA: azacitidine; VEN: venetoclax; LDAC: low-dose cytarabine; CTX: chemotherapy, GO: gentuzumab-ozogamicin; IVO: ivosidenib; inmunoth.: immunotherapy; mut: mutated.

**Table 1 cancers-14-04756-t001:** Summary of main trials with new targets and therapies for acute myeloid leukemia.

		Autor (Reference)	Design (Phase/Patients)	Targeted-Therapy	Efficacy
**First-line for “fit” patients**	Gemtuzumab ozogamicin (GO)	Castaigne S et al., 2012 [7]	CTX + GO vs. placebo (phase III/280)	CD33+ AML	2-year OS: 53.2% vs. 41.9% (*p* = 0.037) RFS: 50.3% vs. 22.7% (*p* = 0.0003) 2-year EFS: 40.8% vs. 17.1% (*p* = 0.0003)
	Vyxeos (CPX-351)	Lancet JE et al., 2018 [15]	Vyxeos vs. 7 + 3 induction (phase III/309)	s-AML, t-AML, AML-MRC	OS: 9.5 vs. 5.9 m (*p* = 0.003) ORR: 47.7% vs. 33.3% (*p* = 0.016)
	Midostaurin	Stone RM, et al., 2017 [18]	CTX + midostaurin vs. placebo (phase III/3277)	FLT3-mut AML	4-year OS: 51.4% vs. 44.3% (*p* = 0.009) EFS: HR 0.78 (*p* = 0.002)
	Quizartinib	Erba H, et al., 2022 [22]	CTX + quizartinib vs. placebo (phase 3/3468)	FLT3-mut AML	OS: 31.9 vs. 15.1 m (HR 0.77; 95% CI 0.61–0.97) RFS: HR 0.733; 95% CI 0.55–0.97
	Venetoclax	Chua CC, et al., 2020 [23]	Venetoclax + 5 + 2 induction (phase Ib/51)	Bcl-2 inhibitor	ORR: 72% (97% de novo, 43% secondary AML)
	Ivosidenib/ enasidenib	Stein EM, et al., 2021 [24]	CTX + ivo/ena (phase 1/60–91)	IDH1/IDH2-mut AML	ORR/CR: 77%/55% (ivosidenib), 63%/47% (enasidenib)
**First-line for “unfit” patients**	Venetoclax	DiNardo CD, et al., 2020 [29]	AZA + venetoclax vs. placebo (phase III/431)	Bcl-2 inhibitor	OS: 14.7 vs. 9.6 m (*p* < 0.001) CR: 36.7 vs. 17.9% (*p* < 0.001)
		Wei AH, et al., 2020 [32]	LDAC + venetoclax vs. placebo (phase III/211)		OS: 8.4 vs. 4.1 m (*p* = 0.04) CR + CRi: 48% vs. 13%
	Glasdegib	Cortes JE, et al., 2019 [35]	LDAC + glasdegib (phase II/132)	Hedgehog inhibitor	OS: 8.8 vs. 4.9 m (*p* = 0.0004) CR: 17% vs. 2.3%
	Ivosidenib	Montesinos P, et al., 2022 [37]	AZA+ ivosidenib vs. placebo (phase III/146)	IDH-1 mut AML	OS: 24 vs. 7.9 m (*p* = 0.001) 12-months EFS: 38% vs. 12% (*p* = 0.002)
	Magrolimab	Daver NG, et al., 2022 [40]	Magrolimab + AZA (phase Ib/72)	TP53-mut AML	OS: 10.8 m; median duration of CR 7.7 m. ORR: 48.6% (CR 33.3%)
	APR-246 (Eprenetapopt)	Sallman DA, et al., 2021 [43]	APR-246 + AZA (phase Ib-II/55)	TP53-mut AML	ORR: 64% (CR 36%) OS: 10.8 m.
	Quizartinib	Bergua-Burgues JM, et al., 2022 [44]	AZA/LDAC + venetoclax + quizartinib (phase 1-2/45)	TKI-inhibitor	ORR: 54% Infections (*n* = 35)
	Satabolimab	Brunner AM, et al., 2021 [68]	Sabatolimab + HMAs (phase I/48)	Anti-TIM3 antibody	ORR: 41.2% 12 m-PFS rate: 44%
**Maintenance treatment**	CC-486	Wei AH, et al., 2020 [47]	CC-486 vs. placebo (phase III/472)	Oral azacitidine	OS: 24.7 vs. 14.8 m (*p* < 0.001) RFS: 10.2 vs. 4.8 m (*p* < 0.001)
		De Lima, et al., 2018 [49]	CC-486 as maintenance after allo-HCT (phase II/30)		Acute GHVD (10%), chronic GVHD (30%). 1-year EFS: 86%
	Sorafenib	Burchert A, et al., 2020 [51]	Sorafenib vs. placebo (phase II/83)	FLT3-mut AML	HR relapse or death 0.39 (95% CI 0.18–0.85; *p* = 0.013) 2-year RFS 85% vs. 53.3% (*p* = 0.002)
**Refractory and relapsed AML**	Gilteritinib	Perl AE, et al., 2019 [55]	Gilteritinib vs. salvage CXT (phase III/371)	FLT3-mut AML	OS: 9.3 vs. 5.6 m (*p* < 0.001) EFS: 2.8 vs. 0.7 m (HR 0.79; 95% CI 0.58–1.09) CR/CRi: 34% vs. 15.4%
	Enasidenib/ Ivosidenib	IDHENTIFY study (NCT02577406)	Enasidenib vs. salvage CXT (phase III)	IDH-1 mut AML	Failed to meet the primary endpoint (OS)
	Idasanutlin	Konopleva MY, et al., 2022 [61]	Cytarabine + idasanutlin (phase III/447)	MDM2 inhibitor	Failed to meet the primary endpoint (OS) ORR (38.8% vs. 22%); OR (2.25 95% CI 1.36–3.72)
	Emavusertib (CA-4948)	García-Manero G, et al., 2022 [62]	Emavusertib in monotherapy (phase Ia/49)	IRAK-4 inhibitor	40% CR/CRi (AML with spliceosome mutations).
	Uproleselan	DeAngelo DJ, et al., 2022 [63]	Uproleselan + MEC (phase 1-2/47)	E-selectin antagonist	CR/CRi 41% (CR 35%) OS: 8.8 m
	MENIN inhibitors	Miao H, et al., 2020 [65]	NCT04067336 NCT04811560	KMT2a rearranged and NPM1-mut AML	
	Flotetuzumab (bispecific antibody)	Uy GL, et al., 2021 [69]	Flotetuzumab in monotherapy (phase 1-2/88)	CD123xCD3	ORR: 30% (CR 26.7%) OS: 10.2 m.

AML: acute myeloblastic leukemia; CTX: chemotherapy; m: months; OS: overall survival; RFS: relapse-free survival; EFS: event-free survival; ORR: overall response rate; CR: complete response; CRi: complete response with incomplete hematologic recovery; GO: gemtuzumab ozogamicin; AZA: azacitidine; LDAC: low-doses Ara-C; GVHD: graf-versus-host disease. s-AML: secondary AML, t-AML: therapy-related AML, AML-MRC: AML with myelodysplastia-related changes.

## Data Availability

All data can be found in the text.

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
