# Peer review of "Updates on the Management of Acute Myeloid Leukemia"

_cancers, 2022, doi:10.3390/cancers14194756_

Round 1

Reviewer 1 Report

A survey such as this is difficult since the listing of drugs is extensive and the authors have, it is obvious, tried to be all-inclusive.  Because of that, the review sometimes comes across as a bit superficial.  However, I think it is overall a good review, introducing the readers to the newer/newest drugs available and dividing their treatment of them to First-line "fit" and "unfit" patients with AML; maintenance; and refractory/relapsed patients. 

Minor typographical corrections: 

page 4, line 175:  correct BLC2 to BCL2

page 4, line 185:  delete "to"

page 4, line 191:  add "it" at beginning of line

page 5, line 232: not everyone will understand C5D1 as meaning cycle5, day 1.  Might need to spell it out. 

page 7, line 294:  should be "or" rather that "o" as in "grade 3 or"

page 7, line 296:  by "posterior", I am assuming is meant retrospective?

page 7, line 329:  should be "not" instead of "no" 

The authors are correct in saying that there is very limited clinical data for CAR-T cells.  However, I think that a discussion of the difficulties had with this therapy because of the lack of a universal target antigen should be mentioned and perhaps a brief discussion can be had also of the attempt to identify alternate targets besides CD33 and CD123.  Also, when CAR T was combined with azacitidine, there was an increase in CD70 antigens on the cancer cell surface, allowing the CAR T cells to have a more effective target.  Work done at MGH and published in Cancer Cell  also highlighted elimination of a proteinase that permits the CAR T cell to bind more tightly to the AML cell and kill it more effectively.  A discussion such as this can highlight some of the problems encountered by researchers on CAR T therapy.

Author Response

Dear reviewer.

Thank you very much for all your suggestions.

  1. Minor typographical corrections: 

We have corrected all the minor typographical errors as planned.

  1. CAR-T cells.

We agree with your observation about CAR-T cell treatment. Accordingly, we have expanded a little bit the information about this therapy talking about the problems of current targets, and including some new targets for AML, but clinical data on this therapy are currently scarce: “The major limitation of these targets is hematologic toxicity, which requires consolidation with alloSCT in most cases. Nevertheless, new targets like CLL-1, CD70, NKG2D, more specific to LSC than HSC are being explored”.

In the other hand, although we agree that the work of Leick et al. published in Cancer Cell provides an interesting information on new strategies to improve outcomes with this this type of therapy, given that our review has a completely clinical focus, we think it is a little bit outside the scope of this review.

Thank you for your comments because they have been very helpful. Kind regards.

Reviewer 2 Report

Specific comments:

1.      The title is misleading as most of the contents cover well-known targets and treatment over the last 5-10 years. Hence, a more appropriate title is “updates on the management of acute myeloid leukemia”. If the authors are to keep “new targets and therapies”, the account needs to be completely re-written with pipeline products being discussed.

2.      A lot of abbreviations are used. Please state the complete name with an abbreviation when it first appears in text.

3.      The author describes “fit” and “unfit” AML. Please elaborate on the definition of how one should assess fitness in the current era and what tools should be used. It is now very common for physicians to set an age to determine if patients should receive intensive chemotherapy (especially CBF AML and NPM1+ AML). Such a strategy deprives some older patients of the chance of curative induction chemotherapy. Hence, it is important to state clearly how “fit” and “unfit” states are defined.

4.      Explain what toxicities are associated with gemtuzumab ozogamicin (GO) and the methods use to mitigate these problems.

5.      The ALFA-0701 study also showed benefits in event free survival and overall survival in the favourable and intermediate risk categories. A discussion of these findings is needed.

6.      Can the authors shed light on the minimum number of doses of GO that can be or should be given with “7+3” induction? The UK MRC data in CBF AML showed that one dose of GO with induction achieved excellent survivals.

7.      For CPX-351, the author mentioned MRD < 10-3 was achieved in 57% of patients in complete response. Can the authors elaborate on the marker that was used for MRD assessment in this group of patients with t-AML or AML? Was the MRD estimated based on a particular mutant gene?

8.      Please give details on the RADIUS and RADIUS-X studies for midostaurin. What was the OS benefit when midostaurin was used as maintenance?

9.      For the QuANTUM-First study, did the investigators exclude patients with FLT3 tyrosine kinase domain mutation? If so, please state.

10.  For the CAVEAT study, kindly detail what “5+2” was. There two possible regimens: Daunorubin/cytarabine or Daunorubicin/VP-16. Please give the CR and CRi rates as well instead of just ORR.

11.  The roadmap is over-ambitious, especially with relapsed/refractory AML. Please discuss what realistically might be achieved in the next five years.

Author Response

Dear reviewer.

Thank you very much for all your suggestions. We are convinced that they will enrich the paper and make it more comprehensive and of higher scientific quality. We have made all the suggested changes, as detailed below.

1. The title is misleading as most of the contents cover well-known targets and treatment over the last 5-10 years. Hence, a more appropriate title is “updates on the management of acute myeloid leukemia”. If the authors are to keep “new targets and therapies”, the account needs to be completely re-written with pipeline products being discussed.

We definitely agree with your comment, so we have changed the title as you suggested.

2. A lot of abbreviations are used. Please state the complete name with an abbreviation when it first appears in text.

We have rechecked that all abbreviations are clarified with full name at their first appearance.

3. The author describes “fit” and “unfit” AML. Please elaborate on the definition of how one should assess fitness in the current era and what tools should be used. It is now very common for physicians to set an age to determine if patients should receive intensive chemotherapy (especially CBF AML and NPM1+ AML). Such a strategy deprives some older patients of the chance of curative induction chemotherapy. Hence, it is important to state clearly how “fit” and “unfit” states are defined.

Thank you very much for this comment. This would probably be a topic for a dedicated review, but you are absolutely right that we should include a clarification on this topic. Therefore, we have included a paragraph dedicated to this point in the introduction that highlights the difficulties in this fit/unfit decision and refers the reader to other publications that can help them on this topic.

“One of the most important challenges in the treatment of AML lies in assessing patient fitness to receive intensive treatments, therefore with curative intent. This evaluation should include an integral evaluation of the patient, with clinical characteristics (age, performance status, comorbidities, quality of life, patient’s preferences and family support) evaluable through different geriatric scores and biological factors of the disease. Intensive chemotherapy with attenuated doses can be considered in frail patients with a high probability of long-term remission, high chemosensitivity and a favorable cytogenetic/molecular risk2,4

4. Explain what toxicities are associated with gemtuzumab ozogamicin (GO) and the methods use to mitigate these problems.

Thank you very much for your comment. Gemtuzumab toxicities are detailed in line 99 of the text, where we have included other strategies to mitigate toxicities such as Defibrotide prior to allogeneic transplantation. “The main adverse events are veno-occlusive disease (VOD) and thrombocytopenia. However, the most recent trials did not suggest that the VOD risk is higher than the benefit of GO if patients are selected and the GO unique dose is lower than 6mg/m26. Prophylaxis with defibrotide in alloSCT after GO could be considered”

5. The ALFA-0701 study also showed benefits in event free survival and overall survival in the favourable and intermediate risk categories. A discussion of these findings is needed.

Thank you very much for your comment. To clarify this point, we have highlighted how the ALFA-0701 study despite having some positive results from the addition of GO to conventional chemotherapy in terms of EFS and OS, these results were not seen in high risk patients. Line 76: “The addition of GO to induction and consolidation achieved a significant improvement of event-free survival (EFS) and overall-survival (OS) (2-year OS 53.2% vs 41.9% p=0.037) with no increased in fatal toxicities. Nevertheless, the benefit associated with GO was not apparent in patients with unfavorable cytogenetics (HR 0.59 (0.32-1.09), p=0.31)”.

These data were later confirmed in the meta-analysis by Hills et al. Line 93: Hills et al. confirmed in a later meta-analysis the results of the ALFA-0701 trial and AML15 study. The benefit of GO depended predominantly on those patients with fa-vorable (absolute survival benefit at 6 years 20.7% OR 0.47 (0.31-0.73) p=0.0006) and intermediate cytogenetics (absolute survival benefit at 6 years 5.7%, OR 0.84 (0.75-0.95) p=0.005) while patients with adverse cytogenetic characteristics did not benefit (2.2%, OR 0.99 (0.83-1.18) p=0.9)

6. Can the authors shed light on the minimum number of doses of GO that can be or should be given with “7+3” induction? The UK MRC data in CBF AML showed that one dose of GO with induction achieved excellent survivals.

The ALFA-0701 study that led to the approval of GO in combination with standard chemotherapy for patients with CD33-expressing AML recommends administering GO on days +1, +4 and +7. However, we agree that works such as the one mentioned from the UK`s Medical Research Council evidence efficacy of GO with lower doses and many working groups have adapted their protocols to less intensive schedules. We have therefore included the study, as you suggested.

Line 81: “In the phase 3 AML15 study by the UK`s Medical Research Council, a significant survival benefit was associated with a single dose of GO (3mg/m2) during the first and third courses of chemotherapy in patients with favorable cytogenetics (OS 79% vs 51%; p=0.001)”

Line 105: “To sum up, the combination of GO with induction and consolidation chemotherapy is safe and improves outcomes in patients with de novo CD33+ AML , when administered in fractioned doses of 3mg/m2 (days 1, 4, and 7) or alternatively as a single dose (3mg/m2) on day 1”

7. For CPX-351, the author mentioned MRD < 10-3 was achieved in 57% of patients in complete response. Can the authors elaborate on the marker that was used for MRD assessment in this group of patients with t-AML or AML? Was the MRD estimated based on a particular mutant gene?

Thank you very much for the comment. The phase III study that led to the approval of vyxeos did not include data on minimal residual disease. Some real-life studies, such as this one conducted by the French group, have included MRD data. However, it is real life, and the technique for MRD assessment is variable depending on the technique used at each center. On line 123 you can see that we have made such a clarification. “(MRD was assessed by flow cytometry, NGS (next-generation sequencing) or real-time quantitative polymerase chain reaction according the standard MRD methodology used at site. All patients with MRD <10-3 were considered as negative).”

8. Please give details on the RADIUS and RADIUS-X studies for midostaurin. What was the OS benefit when midostaurin was used as maintenance?

Thank you very much for your comment. Despite the differences observed in the Radius trial in terms of survival (76% vs 85%) these were not statistically significant. However, following your recommendation we have included the data. Line 153: “The estimated 18-month RFS was 89% (69-96%) in the midostaurin arm and 76% (54-88%) in the standard-of-care arm (HR 0.46 (95% CI (0.12-1.86) p=0.27). Despite an improvement in OS with the addition of midostaurin (estimated 24-month OS (95% CI) 85% (65%-94%) versus 76% (54%-89%) (HR, 0.58 [95% CI, 0.19–1.79]; P = 0.34), the results were not statistically significant)”

9. For the QuANTUM-First study, did the investigators exclude patients with FLT3 tyrosine kinase domain mutation? If so, please state.

We mentioned in line 182 how the study included patients with FLT3-ITD mutations. However, following your suggestion we have clarified that patients with FLT3-TKD mutations were excluded from the study. Line 182: “(FLT3-TKD mutations were excluded)”

10. For the CAVEAT study, kindly detail what “5+2” was. There two possible regimens: Daunorubin/cytarabine or Daunorubicin/VP-16. Please give the CR and CRi rates as well instead of just ORR.

As suggested, we have included what the 2+5 scheme consists of, as well as the CR and CRi data from the CAVEAT study.

11. The roadmap is over-ambitious, especially with relapsed/refractory AML. Please discuss what realistically might be achieved in the next five years.

Thank you very much for your comment. We agree that the roadmap is extremely ambitious. However, we want to reflect the possible options for the future treatment of AML. To try to be a bit more realistic, we have highlighted in blue the therapeutic options with positive phase III results, which are likely to represent the present in some countries and the near future in others.

We reiterate our appreciation with the comments you have made to our review which have undoubtedly improved the quality of the review. Thank you

Kind regards

Reviewer 3 Report

This is a run on sentence and too long. Line 23-26. Break in up, her is suggestion-

It is a heterogeneous disease, defined by a broad spectrum of cytogenetic and molecular aberrations  (PUT A PERIOD/FULSTOP HERE). THESE ABERRATIONS are used to divide patients into prognostic subgroups (ELN 2022) AND are conveniently potential therapeutic targets AS WELL.

Line 29-31.

Nevertheless, primary refractory, relapsed, high-risk, and unfit for allogenic stem cell transplantation (alloSCT) patients still do not have effective therapeutic alternatives, HENCE many investigational drugs are currently being explored .

Line 33 please remove the word “landscape”, not appropriate here rather say

The objective of this review is to DISCUSS AML treatment in first-

MAKE THIS  (LINE 38-42) JUST ONE PARAGRAPH NOT 2 PARAGRAPHS

For many years, the only therapeutic alternative for newly diagnosed AML “fit” patients was intensive chemotherapy based on cytarabine + anthracycline (“7+3”). In addition, allogeneic stem cell transplantation played an important role to improve the survival of eligible intermediate or high-risk AML patients. Targeted therapies combined with standard chemotherapy or new formulations of older drugs have expanded the treatment options and improved survival .

LINE 61, CHANGE “AS” TO “SINCE”

SINCE the association between high CD33 expression and the NPM1 mutation is well

LINE 111-112

This is a long run on sentence.

“Its approval was based on the RATIFY trial; patients were randomly assigned to receive induction and consolidation standard therapy plus either midostaurin or placebo and those who were in remission after consolidation therapy entered a maintenance phase with midostaurin or placebo

Line 126

You say “on the other hand” not “in the other hand”

Line 185 -Remove the “to” in  failed to ven +HMA

And say “who failed VEN + HMA display high-risk disease biology”

Line 281

You don’t day “molecular landscape”, preferable say “molecular pathophysiology”

Line 294 Something is missing in this sentence , maybe you intended to say “or”

Noticeably, the adverse events of grade 3 o higher were less frequent in the gilteritinib group .

Line 384

Please stop using the word “landscape”

Line 386

Instead of “in the same way”, say “in the same token”

Author Response

Dear reviewer.

Thank you very much for all your suggestions. We have updated the text following all your suggestions. It has been very helpful.

Kind regards